# Beyond Information Distortion: Imaging Variable-Length Time Series Data for Classification

**DOI:** 10.3390/s25030621

**Published:** 2025-01-21

**Authors:** Hyeonsu Lee, Dongmin Shin

**Affiliations:** Department of Industrial and Management Engineering, Hanyang University, Ansan 15588, Republic of Korea; hyeonsulee@hanyang.ac.kr

**Keywords:** time series classification (TSC), variable-length time series

## Abstract

Time series data are prevalent in diverse fields such as manufacturing and sensor-based human activity recognition. In real-world applications, these data are often collected with variable sample lengths, which can pose challenges for classification models that typically require fixed-length inputs. Existing approaches either employ models designed to handle variable input sizes or standardize sample lengths before applying models; however, we contend that these approaches may compromise data integrity and ultimately reduce model performance. To address this issue, we propose Time series Into Pixels (TIP), an intuitive yet strong method that maps each time series data point into a pixel in 2D representation, where the vertical axis represents time steps and the horizontal axis captures the value at each timestamp. To evaluate our representation without relying on a powerful vision model as a backbone, we employ a straightforward LeNet-like 2D CNN model. Through extensive evaluations against 10 baseline models across 11 real-world benchmarks, TIP achieves 2–5% higher accuracy and 10–25% higher macro average precision. We also demonstrate that TIP performs comparably on complex multivariate data, with ablation studies underscoring the potential hazard of length normalization techniques in variable-length scenarios. We believe this method provides a significant advancement for handling variable-length time series data in real-world applications. The code is publicly available.

## 1. Introduction

Time series data are ubiquitous and appear across various domains, from trajectories [1] and sensor-based human action recognition [2] to monitoring data from facilities and machines in manufacturing systems [3]. Time series data collected from the real-world often have heterogeneous traits, such as *variable length*—the lengths of the series sample are diverse [4]. Given the prevalence of variable-length time series data in real-world scenarios [5], finding effective ways to handle such data is crucial. Previous studies have preprocessed variable-length time series data to make their lengths uniform by sliding a fixed-size window [6], resampling, or introducing artificial values [7]. However, the best method for managing variable-length time series data and their effectiveness are still unclear, with many recent studies having highlighted it as a future research question in TSC [4,8,9,10].

This issue is even more pronounced for time series data collected by sensors, one of the most widely used tools to gather real-world time series data [11]. Such data often contain consistent noise [12], thus making it critical to capture *key local patterns*—the patterns or subsequences that best explain inter-class variability [13] (e.g., time series data from walking versus sitting classes, or normal versus anomalous data in machine monitoring, may each have distinct key local patterns). To capture key local patterns for TSC, 1D-CNN methods are traditionally employed due to their success in detecting local patterns [14]. One-dimensional CNNs apply filters along the time dimension, enabling the network to identify local patterns that distinguish between classes. Additionally, CNNs can detect features regardless of their position within the time series, which is particularly advantageous in TSC, where the location of key local patterns can vary across instances [15,16,17].

However, capturing key local patterns in variable-length time series presents a significant challenge for 1D CNNs. These models require a fixed input size for batch processing, necessitating length uniformization techniques such as truncation, interpolation, or padding. While these methods enable the use of a 1D CNN architecture, they can introduce noise and distort critical information, including key local patterns, in the original data. This may negatively affect model performance by interfering with the capture of key local patterns, eventually leading to suboptimal results. Figure 1 illustrates this phenomenon.

Although alternatives to 1D CNNs are viable for handling data heterogeneity and are frequently used in various domains [4], their ability to capture features for TSC is questionable. Specifically, for non-deep learning-based methods, the separation of feature extraction and downstream tasks can constrain their performance. While RNN-based and transformer-based approaches can preserve data integrity by using mask layers to avoid padding effects, their processing architecture may struggle to capture the often position-variant key local patterns in TSC. This creates an unresolved trade-off between capturing key local patterns and preserving the data integrity of variable-length time series.

Motivated by these limitations, we focus on one primary question: “***How can we effectively capture the key local pattern for TSC while preserving the original representation of variable-length time series?***”

To address this question, we introduce **TIP** (Time series Into Pixels), which directly maps each value at each time step into a pixel in a 2D image, where the vertical axis represents time steps and the horizontal axis represents the value at each time step. By converting 1D time series data into a 2D format, TIP enables the use of CNNs to capture key local patterns while preserving data integrity for variable-length TSC.

To evaluate our method, we conduct extensive experiments, comparing TIP to 10 baseline methods across 11 real-world benchmarks. Without relying on a powerful vision model as a backbone, we use a straightforward LeNet-like 2D CNN model. Through these experiments, TIP consistently outperforms existing methods, showing 2–5% higher accuracy and 10–25% higher macro average precision, demonstrating the effectiveness of our method for variable-length time series classification. Furthermore, we show that TIP performs comparably in complex multivariate scenarios. Finally, through an ablation study, we confirm that length uniformization techniques for handling variable-length data can lead to performance degradation.

Our contributions are twofold:We introduce TIP, a simple yet robust method that converts variable-length time series into a 2D pixel-based format, mapping time steps to the vertical axis and values to the horizontal. This representation enables vision models, including CNNs, to capture key local patterns effectively while preserving data integrity.We validate TIP by employing a straightforward LeNet-like CNN, demonstrating that its performance gains arise from the data representation itself rather than the complexity or power of the vision model.

## 2. Existing Solutions for Variable-Length TSC

In this section, we review existing methods and potential solutions not yet explored in the literature for variable-length TSC. We categorize these solutions into three groups: non-deep learning-based solutions, masking-based solutions, and 1D-CNN-based solutions with preprocessing. We exclude single-batch-sized solutions from deep learning-based methods; although technically feasible, they are impractical due to high variance in batch loss, resulting in unstable and prolonged training. Additionally, this approach underutilizes GPU resources. We organized the main differences between previous methods and our method in Table 1.

We include early classification methods within the scope of non-deep learning-based solutions. Although they could fit other categories depending on the classifier, recent studies employ SVMs, so we include them in this category.

### 2.1. Non-Deep Learning-Based Solutions

Before the emergence of deep learning in TSC, non-deep learning or machine learning models were primarily used. These models configured the feature space by measuring distances between samples or identifying key local patterns or shapelets essential for classification. With flexible feature extraction approaches, many of these models do not require fixed-size inputs, making them suitable for variable-length TSC problems. For example, DTW-1NN [18], a go-to method in TSC [19], handles variable-length time series data using dynamic time warping (DTW) distance, capturing time series characteristics like variations in speed and duration without modifying the series. Similarly, shapelet-based methods extract key local patterns by sampling subsequences or shapelets and selecting the ones that best characterize each class, supporting variable-length TSC without uniform length conversion. Early classification methods further address variable-length TSC by making confident predictions on partial data through adaptive thresholding or window-based segmentation, thus enabling efficient classification without full sequence input.

While these methods are effective, with some performing comparably to state-of-the-art models [8], they deviate from deep learning’s end-to-end approach. The separation between feature extraction and classification can limit their ability to optimize the feature space for identifying key local patterns.

### 2.2. Masking-Based Solutions

RNN-based models were initially the primary choice in deep learning for time series analysis due to their ability to capture temporal dependencies, making them widely used in TSC across various domains. More recently, transformers have gained popularity in time series analysis, particularly for handling variable-length sequences. Having proven their success in NLP, where sentence lengths vary, they manage sequence variability with padding, masking, and selective computation. RNNs handle padded sequences by excluding artificial values from loss calculations through element-wise summation, while transformers use attention mechanisms, assigning large negative values to padding elements, which reduces their softmax attention scores to zero.

While RNNs and transformers perform well in time series tasks like long-term forecasting and imputation, they have limited application in TSC. Apart from other issues detailed in [14], one key limitation is their tendency to capture global dependencies rather than local ones. This poses challenges in identifying key local patterns essential for classification, especially given common variations in these patterns (e.g., phase, warping, offset) in time series data [20,21]. Some approaches utilize RNNs to address variable-length TSC by embedding series into fixed-size latent representations and applying a classifier, similar to a seq2seq architecture [22]. However, this approach underperforms due to ineffective feature extraction, especially when compared with non-deep learning methods and 1D CNN-based deep learning methods, which will be discussed next.

### 2.3. 1D CNN-Based with Preprocessing Solutions

One-dimensional CNN-based methods have become a gold standard in TSC thanks to their robust feature extraction capabilities [19]. By employing 1D convolutional filters, these models effectively capture various local patterns—including phase shifts, warping, and offsets—regardless of where they appear along the temporal axis. Despite these strengths, 1D CNNs typically require fixed-length inputs, posing challenges for TSC tasks with variable-length sequences.

A common workaround is to uniformly extend each time series in a dataset to the maximum sequence length via padding. Unfortunately, this practice can degrade data integrity by mixing valid observations with synthetic zeros, thereby obscuring the true signal. In 1D CNNs, convolutional filters then process both real and padded values together, generating noisy feature maps that struggle to differentiate between authentic and invalid data.

This issue is compounded by z-normalization, a widely recognized preprocessing step in TSC that scales each time series to zero mean and unit variance [19,23,24]. Used extensively across the TSC domain—including many state-of-the-art methods such as InceptionTime [25]—z-normalization promotes stable training and consistent input scaling. However, once normalized, zero-padding can become statistically indistinguishable from genuinely low-value data points, making it even harder for CNN filters to separate signals from padding. This often leads to uniform, non-informative activations in the padded regions, ultimately diluting or biasing the model’s learned features.

While z-normalization helps stabilize training by standardizing each series’ distribution, it can inadvertently cause padded zeros to appear as legitimate (low-valued) data. Consequently, in variable-length TSC settings—where padding is unavoidable—the model may treat spurious regions as meaningful, obscuring true temporal patterns and impairing overall classification performance.

In certain controlled scenarios—such as when zero is a valid and meaningful measurement that naturally appears in the data—padding with zeros may still convey some structural information about sequence length. However, if zero values are frequent or hold no meaningful semantics in a given domain, padding risks merging authentic signals with artificially introduced noise, hindering the network’s ability to differentiate between real observations and placeholders.

These findings highlight the need for careful consideration when applying padding and z-normalization together, especially for sequences of differing lengths. Despite the recognized benefits of z-normalization in accelerating convergence and promoting stable training, practitioners should remain mindful that zero-padding can confound learned feature representations—ultimately hindering the performance of 1D CNN-based TSC models such as InceptionTime [25]. See Figure 2.

In addition to padding, other length uniformization techniques, such as interpolation and truncation, distort the data’s inherent properties (Figure 1a,b). Interpolation can alter event timing within the series, and truncation may remove essential information. Although these techniques enable deep learning models to handle variable-length TSC, they compromise the integrity of the original time series representation, limiting the model’s ability to capture critical characteristics.

While some modified 1D CNN architectures attempt to handle variable-length inputs through masking layers [26,27], these approaches remain limited in scope and are typically evaluated only against older baselines like DTW. Such masking-based strategies assume a certain level of temporal correlation between short and long sequences, which can lead to misrepresentations. In contrast, our method transforms time series data into a 2D representation and applies a two-dimensional CNN (2D-CNN), allowing convolutional models to capture key local patterns with inherent variability while preserving data integrity. Additionally, we compare our method to 10 baselines across 11 real-world benchmark datasets, positioning our approach as an accurate and effective solution for variable-length TSC.

## 3. Proposed Method

### 3.1. Preliminaries: Variable-Length TSC Problem Formulation

Let D={(X,Y)} be our dataset, where $X={\left\{x_{n}^{\left(1:L_{n}\right)}\right\}}_{n=1}^{N}$ consists of variable-length time series data, with each series $x_{n}^{\left(1:L_{n}\right)}:={\left(x_{n}^{t}\right)}_{t=1}^{L_{n}}=\left(x_{n}^{1},\ldots ,x_{n}^{L_{n}}\right)$. Each element $x_{n}^{t}$ within a series lies in $R^{d}$, where $L_{n}$ denotes the length of the *n*-th series and *d* represents the number of variables or variates measured at each time point. We characterize X as a set of variable-length time series if there exist distinct indices $i,j\in \left\{1,\ldots ,N\right\}\subset N^{+}$ such that $L_{i}\neq L_{j}$. For simplicity in notation, we will initially focus on the univariate case (d=1), where $x_{n}^{t}$ is simplified to $x_{n}^{t}\in R$. The primary objective of variable-length TSC is to develop a model that assigns each variable-length times series $x_{n}^{\left(1:L_{n}\right)}$ to one of the possible classes $y_{n}\in Y$.

### 3.2. TIP: Time Series into Pixels

To preserve original data from information distortion in variable-length time series classification (TSC), we introduce a transformation method called *Time series Into Pixels* (*TIP*). In this method, each time series value is mapped to a corresponding pixel using a binary mask: pixels representing the original data values are set to 1, while all other pixels are set to 0. Each time step in the series is mapped to its respective pixel by finding the appropriate horizontal axis index, with the vertical axis representing the time axis.

By transforming the time series into pixels, it eliminates the need to handle invalid values for variable-length time series, thereby preserving the original data representation while possessing the effective feature extraction capabilities of CNNs. The procedure is detailed in Algorithm 1.
**Algorithm 1** TIP: Time series Into Pixels**Input:** Variable-length time series $x_{n}^{\left(1:L_{n}\right)}$**Input:** Maximum length $L_{max}$**Output:** Transformed variable-length time series $I_{n}$**Parameter:** Boundary constants $\beta^{-},\beta^{+}$1:Set the boundary constants to fit the time series data $x_{n}^{\left(1:L_{n}\right)}$ in $\left[\beta^{-},\beta^{+}\right]$2:$I_{n}\leftarrow 0_{L_{max}\times L_{max}}$                     ▹ Initialize image matrix3:**for** each time step column index $i=0,\ldots ,L_{\max}-1$
**do**4:      Find the corresponding row *j* using Equation (Equation 2)5:      Set $I_{n}\left[i,j\right]=1$                 ▹ Mark the corresponding pixel6:**end for**7:**(optional):** For very long sequences, resize $I_{n}$ with certain ratio8:**(optional):** For multivariate case, repeat from step 1 to 6 by each variate and stack9:**return **$I_{n}$

First, we scale the time series values to fit in the image; all time series data $x_{n}^{t}$ for each time step $t\in [1,L_{n}]$ fit to the range $\left[\beta^{-},\beta^{+}\right]$, where $\beta^{-},\beta^{+}$ are boundary constants. This approach ensures that every time series value is effectively mapped to a corresponding position within the 2D representation, preserving the full range of the original data [28,29,30]. Then, we construct an image matrix $I_{n}$ with dimensions $L_{max}\times L_{max}$. Each column of $I_{n}$ corresponds to a time step in the time series, and each row corresponds to a value of $x_{n}^{t}$. To map the time series value to a corresponding row index $j\in \{0,\ldots ,L_{max}-1\}$, the range $\left[\beta^{-},\beta^{+}\right]$ is divided into $L_{\max}$ bins, each covering an interval $\left[\beta^{-}+j\Delta ,\beta^{-}+\left(j+1\right)\Delta \right]$. The incremental amount is calculated as $\Delta =\frac{\beta^{+}-\beta^{-}}{L_{\max}}$. This discretization can be represented as:(1)$$\left[\beta^{-},\beta^{+}\right]\to \left\{\left[\beta^{-},\beta^{-}+\Delta \right],\ldots ,\left[\beta^{-}+j\Delta ,\beta^{-}+\left(j+1\right)\Delta \right],\ldots ,\left[\beta^{+}-\Delta ,\beta^{+}\right]\right\}$$Thus, the value at each time step $x_{n}^{t}$ is mapped to a specific pixel in $I_{n}$ according to:(2)$$x_{n}^{t}\mapsto I_{n}\left[i,j\right]\;\mathrm{where}\;x_{n}^{t}\in \left[\beta^{-}+j\Delta ,\beta^{-}+\left(j+1\right)\Delta \right],\;i=t-1,\;\mathrm{and}\;j\in \left[0,L_{max}-1\right]\subset N_{0}$$

One of the notable strengths of our method is its dynamic adjustment of the representation of the original time series data through the hyperparameters $\beta^{-},\beta^{+}$. Since the incremental value $\Delta =\frac{\beta^{+}-\beta^{-}}{L_{\max}}$ between each discretized interval $\left[\beta^{-}+j\Delta ,\beta^{-}+\left(j+1\right)\Delta \right]=\left[\beta^{-}+j\left(\frac{\beta^{+}-\beta^{-}}{L_{\max}}\right),\beta^{-}+\left(j+1\right)\left(\frac{\beta^{+}-\beta^{-}}{L_{max}}\right)\right]$ is derived from $\beta^{-},\beta^{+}$, a higher value of the range $\left[\beta^{-},\beta^{+}\right]$ results in a wider range for each discretized interval, whereas a lower value means a narrower range. This enables dynamic adjustments to the representation of time series data, allowing models to consider the appropriate scale or shape of it.

Unlike traditional techniques that rely on padding, interpolation, or truncation, our Time series Into Pixels (TIP) method transforms a variable-length time series into a two-dimensional binary pixel representation without distorting the underlying signal. Unlike existing variable-length TSC solutions that typically rely on one-dimensional architectures with masking layers or manual temporal alignment, TIP directly maps each original time series value to a unique pixel, effectively preserving the original timing, scale, and structure of the data. This representation simultaneously enables the use of vision models such as 2D-CNNs, which are known to be effective in capturing local patterns across spatial dimensions. Thus, TIP departs from prior approaches that often either compromise data integrity or require specialized modules to handle variable-length inputs.

### 3.3. 2D-CNN Backbone

To verify the effectiveness of our 2D representation without relying on the performance of a strong backbone model, we employed a very straightforward 2D CNN model similar to LeNet [31], the primitive CNN model. Like LeNet, our model consists of standard convolutional layers, pooling layers, and a fully connected layer. The primary differences between our model and LeNet are the inclusion of dropout layers and a global pooling layer, which are now commonly used to improve model robustness with lower computational cost. See Figure 3.

We used two baselines for TIP: ***Not-Refined TIP*** and ***Refined TIP***. Not-Refined TIP uses the same architecture and hyperparameters (i.e., batch size, learning rate, β=4) across all datasets, hence the name. Refined TIP, on the other hand, involves tuning β and all hyperparameters of the simple backbone model, such as the number of convolutional layers and the units within specific ranges. Despite these refinements, it retains only the basic features of LeNet, including convolutional layers, pooling layers, and dense layers. Detailed explanations of our backbone are provided in Appendix A.

## 4. Experiment

### 4.1. Datasets and Metrics

To evaluate the effectiveness of our method on real-world datasets, we conducted experiments on 11 variable-length time series datasets from the UCR Time Series Classification Archive (https://www.cs.ucr.edu/~eamonn/time_series_data_2018/, accessed on 10 July 2024), one of the most widely used benchmarks in the TSC domain. For evaluation metrics, we used accuracy and macro average precision (MAP) since all datasets are class-balanced. Detailed descriptions of the datasets are available in Appendix A.

### 4.2. Baselines for Comparison

We compared our method with the leading existing solutions for variable-length TSC, as introduced in Section 2. We selected the top-performing methods from each category. Below is a brief description of the methods used in our comparison and how each method handles variable-length inputs:

#### 4.2.1. Non-DL Methods

ED-1NN: A simple distance-based baseline using Euclidean distance for classification. Often used as a reference due to its simplicity. To handle variable-length sequences, ED-1NN typically requires padding or truncation of time series to a fixed length. We chose zero-padding here.DTW-1NN [18]: Long considered the gold standard for TSC, DTW-1NN excels at handling misaligned time series and has remained difficult to surpass [32]. DTW inherently handles variable-length sequences by allowing elastic shifts in the time axis, effectively aligning sequences of different lengths without the need for padding.TEASER [33]: An early classifier that efficiently handles variable-length inputs without extensive preprocessing. TEASER efficiently processes variable-length time series data by identifying critical patterns early in the sequence, without being influenced by the overall sequence length.RDST [34]: A shapelet-based method known for extracting key local patterns, providing one of the state-of-the-art performances on TSC [8]. RDST manages variable-length sequences by identifying and utilizing shapelets of different lengths, allowing it to flexibly match and classify time series without requiring uniform sequence lengths.

#### 4.2.2. Masking Methods

Vanilla RNN [35]: A basic recurrent model using masking to handle variable-length sequences, though surpassed by more advanced architectures. Masking involves padding shorter sequences and using a mask to indicate the actual length, ensuring that the RNN processes only the valid time steps during training and inference.BiLSTM [36]: Bidirectional LSTM, leveraging masking and bidirectional processing, and offering more robust results than RNN. Similar to vanilla RNN, BiLSTM uses masking to manage variable-length inputs, allowing the model to capture dependencies in both forward and backward directions without being affected by the padded values.TST [37]: A transformer-based model using masking, with competitive performance across diverse time series datasets. In TST, masking is employed to handle variable-length sequences by padding shorter sequences and applying attention masks to prevent the model from attending to padded positions, thereby maintaining the integrity of the actual data.

#### 4.2.3. 1D-CNN Methods

FCN [38]: A fully convolutional model that efficiently extracts spatial patterns, widely used for TSC tasks. To handle variable-length sequences, FCN typically applies preprocessing steps such as padding, truncation, or interpolation to convert all input time series to a fixed length.ResNet [38]: A deep architecture adapted from image classification, known for its strong results in TSC [39]. Like FCN, ResNet also manages variable-length time series by incorporating preprocessing steps like padding, truncation, or interpolation to standardize input lengths.InceptionTime [25]: A top-performing deep learning model for TSC, consistently demonstrating superior performance with its multi-scale architecture, as highlighted in recent reviews [8]. We selected this baseline for its robustness, though variants like LITETime [40] offer reduced computation time while maintaining the same performance as the original. InceptionTime handles variable-length inputs by utilizing preprocessing techniques such as padding, truncation, or interpolation to achieve a fixed input size.

As evaluated in a recent review [8], other methods such as [41,42,43] have also demonstrated strong performance in TSC. However, we excluded these methods because, when compared to the baselines we selected, the performance differences were negligible, and they were either computationally too expensive or not applicable to variable-length TSC due to fixed size input constraint.

### 4.3. Implementation Details

Following prior publications [18,25] and the recommendations from the UCR Archive [19,24], all experiments used the default training and test splits provided by the UCR Archive, and each series was z-normalized. We implemented the best architectures reported in the literature, such as using five classifiers in the ensemble for InceptionTime [25] and 10,000 shapelets for RDST [34].

Since few studies adopt RNN-based models for TSC, we developed the vanilla RNN and BiLSTM in-house. We provide the detailed implementations of the two models in Appendix A. In addition to these models, we used implementations available in aeon (https://www.aeon-toolkit.org, accessed on 10 July 2024) and tsai (https://timeseriesai.github.io/tsai, accessed on 10 July 2024), the popular third-party libraries in time series analysis.

For 1D-CNN methods, we applied various length uniformization techniques, including truncation (TC), interpolation (IP), and padding (PD). For the most powerful 1D-CNN-based method, InceptionTime, we applied all three length uniformization techniques, while for other 1D-CNN baselines, we used only padding, as it yielded the best performance.

For all deep learning-based baselines, we added an early-stopping strategy and a plateau learning rate scheduler to avoid overfitting and getting stuck in local optima. These strategies are often omitted in the related literature despite their importance. By implementing these strategies and fine-tuning the batch size and learning rate over sufficient trials, we achieved improved performance for the baselines compared to their reported results. Additional implementation details, including the search space for fine-tuning and training strategy, are provided in Appendix A. All models were trained using the TensorFlow framework on a single NVIDIA RTX 2080 Ti GPU (NVIDIA Corporation, Santa Clara, CA, USA). We have made the complete source code available.

### 4.4. Results

As demonstrated in Figure 4, our approach consistently achieves the best or second-best results in both accuracy and MAP. Specifically, non-DL methods exhibited inferior performance due to the separation between configuring the feature space and the downstream classification task. In particular, ED-1NN recorded significantly lower performance than DTW-1NN because it failed to capture key aspects of variable-length sequences, such as speed variation. Although TEASER and RDST showed relatively decent performance, the aforementioned two-step learning process hindered the optimal configuration of the feature space for classification.

Similarly, recurrent models and transformers also demonstrated subpar performance. While these models maintain data integrity through the use of masking, their classification performance remains low. Beyond the traditional reasons for their low performance in TSC [14], their forward structures are inadequate for classification tasks where key local patterns are often position-variant, making it difficult to effectively capture the underlying dynamics. Although TST outperformed both recurrent models and transformers, it still lagged behind 1D-CNN-based methods and our convolutional model-based method, TIP, which excel at learning local patterns.

Lastly, 1D-CNN-based models generally performed well; however, our method, TIP, outperformed them across most datasets. Notably, InceptionTime, which employs truncation as a preprocessing step, achieved the lowest performance despite being the latest and ostensibly the best-performing algorithm in its category. This result indicates that truncation as a preprocessing method significantly distorts the original data information. When padding, instead of truncation, was applied, InceptionTime, ResNet, and FCN showed progressively higher performance. This improvement can be attributed to their complex structures: InceptionTime utilizes ensemble techniques and inception blocks, which are modern feature extraction methods; ResNet incorporates residual connections; and FCN consists solely of convolutional layers. Additionally, padding generally outperformed interpolation, likely because interpolation excessively alters the temporal dynamics of the original data. However, overall, these imputation methods using artificial values resulted in lower performance compared to our method, which preserves data integrity by employing a straightforward vision model with a 2D-CNN backbone.

## 5. Additional Analysis

### 5.1. What About in Multivariate Setting?

To evaluate the effectiveness of our method in multivariate setting, we conducted experiments on variable-length and multivariate real-world benchmark datasets from the UCI Archive [44]: JapaneseVowels and CharacterTrajectories. For this analysis, each variate time series data was converted into an image and then stacked (i.e., three variates = three-channel images). Note that we continued to utilize the simple 2D CNN as a backbone for this analysis. Unlike other baselines, this model does not capture the relationships between variables, which is crucial in multivariate TSC.

As shown in Table 2, our method achieves performance comparable to other methods, despite lacking features specifically designed to capture spatial patterns between variables. While InceptionTime, the current state-of-the-art TSC integrated with length uniformization techniques, achieved the highest accuracy, our method delivered a comparable accuracy score and outperformed it in MAP. Nevertheless, our approach may face computational challenges when scaling to extremely large datasets or handling high-dimensional multivariate settings. This is primarily because each variate is treated as a separate channel and stacked, which could increase memory and processing demands. However, our results suggest that even under these constraints, the method remains effective in maintaining high performance and robustness for complex multivariate TSC problem.

### 5.2. Is Preserving the Original Representation Really a Matter in Variable-Length TSC?

To investigate the impact of information distortion caused by length uniformization techniques in variable-length TSC, we conducted an ablation study of our method. We assessed performance on the PGWZ dataset using Not-Refined TIP with four different strategies for length uniformization: zero-padding, interpolation, truncation, and no adjustment (Default), which is our approach.

As shown in Table 3, the performance in terms of accuracy and MAP significantly dropped when using truncation, interpolation, and padding compared to no length uniformization techniques. This demonstrates that distorting the representation of time series data has a critical impact on model performance.

## 6. Conclusions and Discussion

This work highlights the challenges inherent in applying current deep learning methods to the variable-length TSC problem. To address these challenges, we propose TIP, a novel approach that directly maps time series data into 2D image representations. We believe our findings contribute significantly to advancing the handling of variable-length time series, which represent the most natural form of time series data encountered in real-world applications.

While our results in variable-length TSC are promising, several issues remain unresolved. One notable concern is the potential for information loss when resizing series, particularly for long sequences—a challenge that is pervasive and yet unresolved in current TSC methods. Furthermore, our method treats each variate as a separate channel, which could pose challenges when dealing with high-dimensional data or sparse datasets. Such scenarios may result in computational inefficiencies or difficulty capturing the complex interdependencies between variates. To address these challenges, future research could explore methods to better preserve interdependencies in multivariate time series. For example, instead of treating each variate as a distinct channel, randomly positioning multivariate images into a shared channel stack could allow the backbone vision model to capture intricate relationships between variables more effectively. Alternatively, incorporating feature extraction or attention mechanisms tailored to multivariate data may enhance both computational efficiency and performance. Resolving these limitations remains an open area for further investigation, which we aim to address in future work.

## Figures and Tables

**Figure 1 sensors-25-00621-f001:**
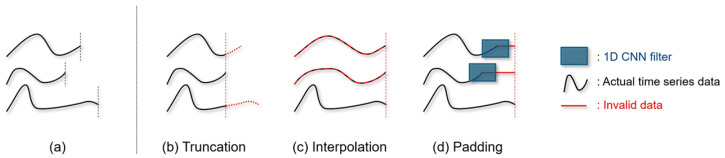
(**a**) Toy example illustrations of variable-length time series data. (**b**–**d**) Common techniques to uniformize the variable-length time series data. (**b**) Truncation cuts off parts of the series to fit a predetermined length, potentially removing crucial information. (**c**) Interpolation inserts data points to adjust the series to a standard length, distorting true temporal dynamics. (**d**) Padding adds repeating values at the end, creating misleading patterns and forcing 1D convolutional filters to process both actual and invalid data indiscriminately.

**Figure 2 sensors-25-00621-f002:**
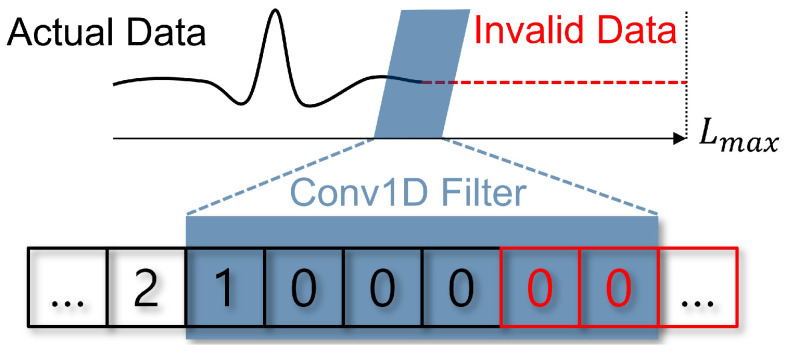
Illustration of the convolution operation with padding in a 1D CNN. Padding extends the series to the maximum length $L_{\max}$, introducing artificial values that can affect the feature maps and the performance of the CNN.

**Figure 3 sensors-25-00621-f003:**
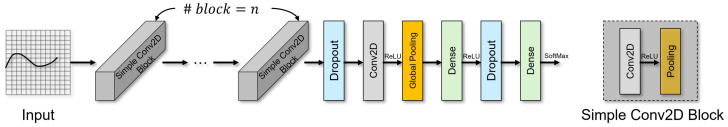
Illustration of our simple 2D CNN backbone. Detailed settings (e.g., kernel size, dense units) are provided in the Appendix A.

**Figure 4 sensors-25-00621-f004:**
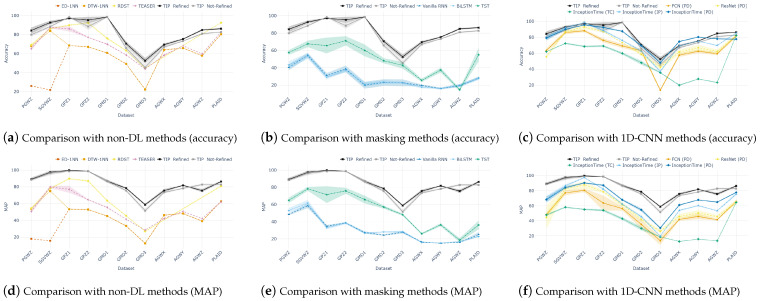
Comparison to existing variable-length TSC solutions on 11 UCR real-world benchmarks, evaluated by (**a**–**c**) Accuracy (%) and (**d**–**f**) Macro Average Precision (MAP), with standard deviations computed across five seeds.

**Table 1 sensors-25-00621-t001:** Comparison of existing solutions and our proposed method for variable-length TSC.

Aspect	Non-Deep Learning-Based (Section 2.1)	Masking-Based (Section 2.2)	1D CNN-Based (Section 2.3)	Ours
End-to-end learning	×	✓	✓	✓
Effective feature extraction for TSC	✓	×	✓	✓
Preserving original data integrity	✓	✓	×	✓

**Table 2 sensors-25-00621-t002:** A performance comparison with existing solutions for variable-length TSC in multivariate settings. **Best results** are highlighted in **bold**. Note that our backbone lacks the ability to capture spatial patterns and the relationships between variables.

Dataset	JapaneseVowels		CharacterTrajectories	
# Variates	12		3	
Lengths	[7,29]		[60,182]	
Train/Test	270/370		1422/1436	
Metrics	Accuracy	MAP	Accuracy	MAP
DTW-1NN	96.93	94.01	66.22	48.46
Vanilla RNN	25.52 ± 0.50	22.44 ± 0.24	22.76 ± 3.48	20.99 ± 0.58
FCN (PD)	95.10 ± 1.36	90.43 ± 2.47	76.43 ± 1.87	61.53 ± 2.11
ResNet (PD)	98.75 ± 0.16	97.44 ± 0.33	78.38 ± 2.03	65.20 ± 1.81
InceptionTime (IP)	99.21 ± 0.03	98.40 ± 0.06	**82.81** ± 0.40	69.28 ± 0.45
InceptionTime (PD)	**99.39** ± 0.05	98.73 ± 0.10	82.43 ± 0.34	68.67 ± 0.55
Not Refined TIP	98.43 ± 0.07	99.66 ± 0.03	55.30 ± 2.99	58.74 ± 2.60
Refined TIP	98.64 ± 0.23	**99.69** ± 0.05	78.11 ± 1.18	**83.35** ± 0.83

**Table 3 sensors-25-00621-t003:** An ablation study comparing three length uniformization techniques—truncation (TC), interpolation (IP), and padding (PD)—and their impact on information distortion in the original representation, as opposed to our technique, which introduces no information distortion.

Strategy	Accuracy	MAP
TIP (TC)	46.80 ± 1.60	58.13 ± 0.91
TIP (IP)	72.80 ± 2.04	82.81 ± 3.40
TIP (PD)	78.00 ± 2.83	85.78 ± 1.96
**TIP (Default)**	**80.00** ± **2.19**	**88.57** ± **0.36**

## Data Availability

The data supporting the reported results in this study are publicly available. All datasets used in this research are accessible from the UCR Time Series Classification Archive at https://www.cs.ucr.edu/~eamonn/time_series_data_2018/ (accessed on 10 July 2024).

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
