# Peer review of "Beyond Information Distortion: Imaging Variable-Length Time Series Data for Classification"

_sensors, 2025, doi:10.3390/s25030621_

Round 1

Reviewer 1 Report

Comments and Suggestions for Authors

This papre studies the classification of time series data, which is a very popular and important issue. Although the length of monitoring data varies, time-series data classification is the classification of the current monitoring data. For this purpose, data is typically generated using a fixed length window and labeled with the last frame category. I think the varialbe input data length in this article is a self created problem, which is unnecessary for data classification. In addition, the author's statement that zero padding in CNN models is non-informative is also inaccurate. The CNN convolution is the process of averaging data, and zero padding does not affect the calculation results, which can make the model faithful the input data. Finally, in the experiment section, the authors should provide a more detailed introduction on how the comparative models achieve variable length data modeling, and provide a more comprehensive analysis of the reasons of the experimental results.

Author Response

Comments 1: Variable input data length in this article is a self-created problem, which is unnecessary for data classification.

Response 1: We appreciate your thoughtful comment highlighting an important perspective on variable-length TSC. We fully understand that it's a common approach to use a sliding window of fixed length and classify sequences based on the label of the last frame when handling the current monitoring time series data. 

The main motivation of our study for addressing variable-length TSC arises from practical constraints observed in real-world applications. During our human-subject studies, we observed that human action sensor data lengths often vary due to intra- and inter-variability among participants, which also reported in [R6].

The issue of variable-lengths occurs in other domains such as manufacturing, as a recent review [R4] reported that more than half of the monitoring datasets in manufacturing processes inherently exhibit variable-length properties. Additionally, variable-length TSC remains a significant challenge in the TSC field, as highlighted earlier [R5] and in recent works [R1]. More significantly, two recent review papers [R2, R3] identify it as an unresolved research question and propose it as a key direction for future work. We authors believe that our study does not deviate from this line of work.

In response to your valuable feedback, we have revised the manuscript to emphasize the prevalence of this problem and added common approaches variable-length, including the fixed size window that you mentioned, and more references in the introduction section of the updated version as follows (These changes can be found in page number 1, 1st paragraph):

“Time series data is ubiquitous and appears across various domains, from trajectories[1] and sensor-based human action recognition[2] to monitoring data from facilities and machines in manufacturing systems[3]. Time series data collected from the real-world often has heterogeneous traits, such as variable-length–the lengths of the series sample are diverse [4]. Given the prevalence of variable-length time series data in real-world scenarios[5], finding effective ways to handle such data is crucial. Previous studies preprocessed the variable-length time series data to make their lengths uniform by sliding a fixed-size window[6], resampling, or introducing artificial values[7]. However, the best method for managing variable-length time series and their effectiveness are still unclear, as many recent studies have illuminated it as a future research question in TSC. [4,8–10]"

  • [R1]: Tan, Chang Wei, et al. "Time series classification for varying length series." arXiv preprint arXiv:1910.04341 (2019).
  • [R2]: Middlehurst, Matthew, Patrick Schäfer, and Anthony Bagnall. "Bake off redux: a review and experimental evaluation of recent time series classification algorithms." Data Mining and Knowledge Discovery (2024): 1-74.
  • [R3]: Wang, Yuxuan, et al. "Deep time series models: A comprehensive survey and benchmark." arXiv preprint arXiv:2407.13278 (2024).
  • [R4]: Farahani, Mojtaba A., et al. "Time-series classification in smart manufacturing systems: An experimental evaluation of state-of-the-art machine learning algorithms." Robotics and Computer-Integrated Manufacturing 91 (2025): 102839.
  • [R5]: Kahveci, Tamer, and Ambuj Singh. "Variable length queries for time series data." Proceedings 17th International Conference on Data Engineering. IEEE, 2001.
  • [R6]: Cervantes, Pablo, et al. "Implicit neural representations for variable length human motion generation." European Conference on Computer Vision. Cham: Springer Nature Switzerland, 2022.

Comments 2: The author's statement that zero padding in CNN models is non-informative is also inaccurate. The CNN convolution is the process of averaging data, and zero padding does not affect the calculation results, which can make the model faithful the input data.

Response 2: We sincerely appreciate your insightful comment on the role of zero-padding in CNN models. We acknowledge your perspective that zero-padding can preserve the integrity of data input for certain 1D-CNN applications. To address this concern, we have elaborated on these nuances and clarified the conditions under which zero-padding may or may not be considered informative in Subsection 2.3 (1D CNN-based Approaches with Preprocessing). We hope this additional context more clearly conveys our position and the specific challenges associated with zero-padding in our particular setting.

“1D CNN-based methods have become a gold standard in TSC thanks to their robust feature extraction capabilities [R3]. By employing 1D convolutional filters, these models effectively capture various local patterns—including phase shifts, warping, and offsets—regardless of where they appear along the temporal axis. Despite these strengths, 1D CNNs typically require fixed-length inputs, posing challenges for TSC tasks with variable-length sequences.

A common workaround is to uniformly extend each time series in a dataset to the maximum sequence length via padding. Unfortunately, this practice can degrade data integrity by mixing valid observations with synthetic zeros, thereby obscuring the true signal. In 1D CNNs, convolutional filters then process both real and padded values together, generating noisy feature maps that struggle to differentiate between authentic and invalid data.

This issue is compounded by z-normalization, a widely recognized preprocessing step in TSC that scales each time series to zero mean and unit variance [R1,R2,R3]. Used extensively across the TSC domain—including many state-of-the-art methods such as InceptionTime [R4]—z-normalization promotes stable training and consistent input scaling. However, once normalized, zero-padding can become statistically indistinguishable from genuinely low-valued data points, making it even harder for CNN filters to separate signal from padding. This often leads to uniform, non-informative activations in the padded regions, ultimately diluting or biasing the model’s learned features.

While z-normalization helps stabilize training by standardizing each series’ distribution, it can inadvertently cause padded zeros to appear as legitimate (low-valued) data. Consequently, in variable-length TSC settings—where padding is unavoidable—the model may treat spurious regions as meaningful, obscuring true temporal patterns and impairing overall classification performance.

In certain controlled scenarios—such as when zero is a valid and meaningful measurement that naturally appears in the data—padding with zeros may still convey some structural information about sequence length. However, if zero values are frequent or hold no meaningful semantics in a given domain, padding risks merging authentic signals with artificially introduced noise, hindering the network’s ability to differentiate between real observations and placeholders.

These findings highlight the need for careful consideration when applying padding and z-normalization together, especially for sequences of differing lengths. Despite the recognized benefits of z-normalization in accelerating convergence and promoting stable training, practitioners should remain mindful that zero-padding can confound learned feature representations—ultimately hindering the performance of 1D CNN-based TSC models such as InceptionTime [R4].”

  • [R1] Ratanamahatana, Chotirat Ann, and Eamonn Keogh. "Three myths about dynamic time warping data mining." Proceedings of the 2005 SIAM international conference on data mining. Society for Industrial and Applied Mathematics, 2005.
  • [R2] Rakthanmanon, Thanawin, et al. "Addressing big data time series: Mining trillions of time series subsequences under dynamic time warping." ACM Transactions on Knowledge Discovery from Data (TKDD)3 (2013): 1-31.
  • [R3] Bagnall, Anthony, et al. "The great time series classification bake off: a review and experimental evaluation of recent algorithmic advances." Data mining and knowledge discovery31 (2017): 606-660.
  • [R4] Ismail Fawaz, Hassan, et al. "Inceptiontime: Finding alexnet for time series classification." Data Mining and Knowledge Discovery6 (2020): 1936-1962.

Comments 3: In the experiment section, the authors should provide a more detailed introduction on how the comparative models achieve variable length data modeling, and a more comprehensive analysis of the reasons of the experimental results.

Response 3: We appreciate your valuable feedback regarding the experimental section of our manuscript. We have carefully reviewed the submitted manuscript and have fixed these issues for our updated version: We added how each comparative model processes variable-length data in Section 4.2 Baselines for comparison (page 7) and more analysis for the experimental results in Section 4.4 Results (page 8 and 9).

- Section 4.2 Baselines for comparison (page 7)

“We compared our method with the leading existing solutions for variable-length TSC,

as introduced in Section 2. We selected the top performing methods from each category.

Below is a brief description of the methods used in our comparison:

Non-DL Methods

  • ED-1NN : A simple distance-based baseline using Euclidean distance for classification.

Often used as a reference due to its simplicity. To handle variable-length sequences,

ED-1NN typically requires padding or truncation of time series to a fixed length. We

chose zero-padding here.

  • DTW-1NN [19]: Long considered the gold standard for TSC, DTW-1NN excels at

handling misaligned time series and has remained difficult to surpass [28, 29]. DTW

inherently handles variable-length sequences by allowing elastic shifts in the time axis,

effectively aligning sequences of different lengths without the need for padding.

  • TEASER [30]: An early classifier that efficiently handles variable-length inputs without

extensive preprocessing. TEASER efficiently processes variable-length time series data by identifying critical patterns early in the sequence, without being influenced by the overall sequence length.

  • RDST [31]: A shapelet-based method known for extracting key local patterns, provid-

ing one of the state-of-the-art performances on TSC [ 8 ]. RDST manages variable-length

sequences by identifying and utilizing shapelets of different lengths, allowing it to

flexibly match and classify time series without requiring uniform sequence lengths.

Masking Methods

  • Vanilla RNN [32]: A basic recurrent model using masking to handle variable-length

sequences, though surpassed by more advanced architectures. Masking involves

padding shorter sequences and using a mask to indicate the actual length, ensuring

that the RNN processes only the valid time steps during training and inference.

  • BiLSTM [33]: Bidirectional LSTM, leveraging masking and bidirectional processing,

offering more robust results than RNN. Similar to Vanilla RNN, BiLSTM uses masking

to manage variable-length inputs, allowing the model to capture dependencies in both

forward and backward directions without being affected by the padded values.

  • TST [34]: A transformer-based model using masking, with competitive performance

across diverse time series datasets. In TST, masking is employed to handle variable-

length sequences by padding shorter sequences and applying attention masks to

prevent the model from attending to padded positions, thereby maintaining the

integrity of the actual data.

1D-CNN Methods

  • FCN [35]: A fully convolutional model that efficiently extracts spatial patterns, widely

used for TSC tasks. To handle variable-length sequences, FCN typically applies

preprocessing steps such as padding, truncation, or interpolation to convert all input

time series to a fixed length.”

  • ResNet [35]: A deep architecture adapted from image classification, known for its

strong results in TSC [36]. Like FCN, ResNet also manages variable-length time series

by incorporating preprocessing steps like padding, truncation, or interpolation to

standardize input lengths.

  • InceptionTime [24]: A top-performing deep learning model for TSC, consistently

demonstrating superior performance with its multi-scale architecture, as highlighted

in recent reviews [ 8 ]. We selected this baseline for its robustness, though variants

like LITETime [ 37] offers reduced computation time while maintaining the same performance as the original. InceptionTime handles variable-length inputs by utilizing

preprocessing techniques such as padding, truncation, or interpolation to achieve a

fixed input size.”

- Section 4.4 Results (page 8 and 9).

“As demonstrated in Figure 4., our approach consistently achieves the best or second-best results in both accuracy and MAP. Specifically, Non-DL methods exhibited inferior performance due to the separation between configuring the feature space and the downstream classification task. In particular, ED-1NN recorded significantly lower performance than DTW-1NN because it failed to capture key aspects of variable-length sequences, such as speed variation. Although TEASER and RDST showed relatively decent performance, the aforementioned two-step learning process hindered the optimal configuration of the feature space for classification.

Similarly, recurrent models and transformers also demonstrated subpar performance. While these models maintain data integrity through the use of masking, their classification performance remains low. Beyond the traditional reasons for their low performance in TSC, their forward structures are inadequate for classification tasks where key local patterns are often position-variant, making it difficult to effectively capture the underlying dynamics. Although TST outperformed both recurrent models and transformers, it still lagged behind 1D-CNN-based methods and our convolutional model-based method, TIP, which excel at learning local patterns.

Lastly, 1D-CNN-based models generally performed well; however, our method, TIP, outperformed them across most datasets. Notably, InceptionTime, which employs truncation as a preprocessing step, achieved the lowest performance despite being the latest and ostensibly the best performing algorithm in its category. This result indicates that truncation as a preprocessing method significantly distorts the original data information. When padding, instead of truncation, was applied, InceptionTime, ResNet, and FCN showed progressively higher performance. This improvement can be attributed to their complex structures: InceptionTime utilizes ensemble techniques and inception blocks, which are modern feature extraction methods; ResNet incorporates residual connections; and FCN consists solely of convolutional layers. Additionally, padding generally outperformed interpolation, likely because interpolation excessively alters the temporal dynamics of the original data. However, overall, these imputation methods using artificial values resulted in lower performance compared to our method, which preserves data integrity by employing a straightforward vision model with a 2D-CNN backbone.”

Reviewer 2 Report

Comments and Suggestions for Authors

The paper proposes Time series Into Pixels (TIP), an intuitive yet strong method that maps each time series data into a pixel in 2D representation, where the vertical axis represents time steps and the horizontal axis captures the value at each timestamp.   To evaluate our representation and avoid benefits from a powerful vision model as a backbone, authors employ a straightforward LeNet-like 2D CNN model.  Through extensive evaluations against 10 baseline models across 11 real-world benchmarks, TIP achieves 2-5% higher accuracy and 10-25% higher macro average precision.  The paper demonstrate that TIP performs comparably on complex multivariate data, with ablation studies underscoring the potential hazard of length-normalization techniques in variable-length scenario.    It provides a significant advancement for handling variable-length time series data in real-world applications. The paper has a potential for publication. However, some  corrections and improvements are needed:

1. The paper lacks sufficient analysis of the relationship between the related work and the new proposed method.

2. The authors should consider motive section to highlight their contributions. The novelty of the proposed method seems insufficient.

3. The authors ought to carefully check the English writing as well as the syntax errors from the beginning to the end. Some minor polishing of grammar would also be good. 

4. The authors should cite some references in the some statements in the paper which can give the background knowledge they concerned.

Comments on the Quality of English Language

The authors ought to carefully check the English writing as well as the syntax errors from the beginning to the end. 

Author Response

Comments 1: The paper lacks sufficient analysis of the relationship between the related work and the new proposed method.

Response 1: We appreciate the reviewer’s feedback on the need for more thorough analysis of the relationship between prior work and our proposed method. In our revised manuscript, we have expanded the discussion in Section 2.3 (1D CNN-based Approaches with Preprocessing) to more explicitly situate our approach within the broader landscape of existing variable-length TSC solutions.

"While some modified 1D CNN architectures attempt to handle variable-length inputs through masking layers [R1, R2], these approaches remain limited in scope and are typically evaluated only against older baselines such as DTW. Such masking-based strategies implicitly assume a certain level of temporal correlation between short and long sequences, which can lead to misrepresentations. In contrast, our method avoids these assumptions by mapping each time point into a pixel, allowing each time series to be processed independently without enforced temporal alignment across samples. Additionally, in regions where no actual data points exist, our approach inherently produces an effect equivalent to masking. This ensures that meaningful time steps are emphasized while areas without valid data are effectively ignored—retaining the advantages of masking-based strategies without explicitly relying on them."

  • [R1]: Sawada, Azusa, et al. "Convolutional neural networks for time-dependent classification of variable-length time series." 2022 International joint conference on neural networks (IJCNN). IEEE, 2022.
  • [R2]: Schneider, Manuel, et al. "An end-to-end machine learning approach with explanation for time series with varying lengths." Neural Computing and Applications 36.13 (2024): 7491-7508.

Comments 2: The authors should consider motive section to highlight their contributions. The novelty of the proposed method seems insufficient.

Response 2: We appreciate the reviewer's feedback for highlighting this concern. In our revised manuscript, we strengthened the novelty claims and provide an expanded discussion of our proposed method more explicitly. The changes can be found in Section 3.2. (TIP: Time series Into Pixels):

"Unlike traditional techniques that rely on padding, interpolation, or truncation, our Time series Into Pixels (TIP) method transforms variable-length time series into a two-dimensional binary pixel representation without distorting the underlying signal. Unlike existing variable-length TSC solutions that typically rely on one-dimensional architectures with masking layers or manual temporal alignment, TIP directly maps each original time series value to a unique pixel, effectively preserving the original timing, scale, and structure of the data. This representation simultaneously enables the use of vision models such as 2D-CNNs which are known to be effective in capturing local patterns across spatial dimensions. Thus, TIP departs from prior approaches that often either compromise data integrity or require specialized modules to handle variable-length inputs."

Comments 3: The authors ought to carefully check the English writing as well as the syntax errors from the beginning to the end. Some minor polishing of grammar would also be good.

Response 3: Thank you for raising the writing quality issues, and we apologize for letting them slip through. We have carefully reviewed the submitted manuscript and have fixed the syntax errors for our updated version. Also, We have arranged for an English language revision to ensure the manuscript is free from grammatical errors.

Comments 4: The authors should cite some references in the some statements in the paper which can give the background knowledge they concerned.

Response 4: We appreciate your valuable feedback. We have incorporated additional references to provide readers with more comprehensive background knowledge and to support the key statements as follows:

  1. “Previous studies preprocessed the variable-length time series data to make their lengths uniform by sliding a fixed-size window[R1], resampling, or introducing artificial values[ 7]. However, the best method for managing variable-length time series and their effectiveness are still unclear, as many recent studies have illuminated it as a future research question in TSC. [4,8–10]” – Section 1. Introduction
  2. “However, capturing key local patterns in variable-length time series presents a significant challenge for 1D CNNs. These models require a fixed input size for batch processing, necessitating length-uniformization techniques such as truncation, interpolation, or padding. While these methods enable the use of a 1D CNN architecture, they can introduce noise and distort critical information, including key local patterns, in the original data. This may negatively affect model performance by interfering with the capture of key local patterns, eventually leading to suboptimal results.” – Section 1. Introduction
  3. “Although alternatives to 1D CNNs are viable for handling data heterogeneity and are frequently used in various domains[4 ], their ability to capture features for TSC is questionable. Specifically, for non-deep-learning-based methods, the separation of feature extraction and downstream tasks can constrain their performance. While RNN-based and transformer-based approaches can preserve data integrity by using mask layers to avoid padding effects, their processing architecture may struggle to capture the often position- variant key local patterns in TSC. This creates an unresolved trade-off between capturing key local patterns and preserving the data integrity of variable-length time series.” – Section 1. Introduction
  4. “While our results in variable-length TSC are promising, several issues remain unaddressed. One concern is the potential for information loss when resizing the series, particularly for long sequences, although this issue is common and unsolved among current TSC methods. [R1, R2]” – Section 6. Conclusion

Reviewer 3 Report

Comments and Suggestions for Authors

-          In the literature review, would benefit from a more in-depth discussion of the gaps these methods fail to address. Please explain why previous solutions are insufficient for preserving data integrity in variable-length time series classification.

-          The author could consider transitions between sections. It could be smoother.

-          In Figure 1, It should clearly indicate which graph corresponds to the "left" and "right" positions.

-          Please add quantitative details about the performance improvements (e.g., accuracy, MAP) achieved with TIP compared to specific baseline methods.

-          The authors mentioned this content “Through rigorous evaluations against 10 baselines across 11 real-world benchmarks, we show TIP consistently outperforms existing methods, with 2-5% higher accuracy and 10-25% greater macro average precision. Using a straightforward LeNet-like CNN, we confirm that TIP’s gains stem from its data representation, not model complexity.” Is it your objective or your result? It is presented in the introduction section as your contribution.

-          Expand on the explanation of the mapping process, particularly how the boundary constants 𝛽−, 𝛽+ are determined. This would make the method more accessible to readers unfamiliar with similar techniques.

-          Please discuss potential limitations or scenarios where the TIP representation might struggle

-          It would be beneficial to compare the 2D-CNN to more advanced architectures to demonstrate the method's robustness.

-          Provide more details about the UCR datasets used, especially those with variable-length characteristics.

-          Please expand on why TIP performs comparably or better than InceptionTime in some cases, particularly for MAP scores. Are there specific characteristics of the datasets that favor TIP?

-          Please elaborate on how TIP could be adapted or extended for multivariate time series with complex interdependencies.

Author Response

Comments 1: In the literature review, would benefit from a more in-depth discussion of the gaps these methods fail to address. Please explain why previous solutions are insufficient for preserving data integrity in variable-length time series classification.

Response 1: We appreciate the reviewer’s valuable feedback on this point. In response, we have expanded our literature review on previous 1D-CNN based approach for variable-length TSC. These revisions can be found in Section 2.3 (1D-CNN based with preprocessing Solutions).

“1D CNN-based methods have become a gold standard in TSC thanks to their robust feature extraction capabilities [R3]. By employing 1D convolutional filters, these models effectively capture various local patterns—including phase shifts, warping, and offsets—regardless of where they appear along the temporal axis. Despite these strengths, 1D CNNs typically require fixed-length inputs, posing challenges for TSC tasks with variable-length sequences.

A common workaround is to uniformly extend each time series in a dataset to the maximum sequence length via padding. Unfortunately, this practice can degrade data integrity by mixing valid observations with synthetic zeros, thereby obscuring the true signal. In 1D CNNs, convolutional filters then process both real and padded values together, generating noisy feature maps that struggle to differentiate between authentic and invalid data.

This issue is compounded by z-normalization, a widely recognized preprocessing step in TSC that scales each time series to zero mean and unit variance [R1,R2,R3]. Used extensively across the TSC domain—including many state-of-the-art methods such as InceptionTime [R4]—z-normalization promotes stable training and consistent input scaling. However, once normalized, zero-padding can become statistically indistinguishable from genuinely low-valued data points, making it even harder for CNN filters to separate signal from padding. This often leads to uniform, non-informative activations in the padded regions, ultimately diluting or biasing the model’s learned features.

While z-normalization helps stabilize training by standardizing each series’ distribution, it can inadvertently cause padded zeros to appear as legitimate (low-valued) data. Consequently, in variable-length TSC settings—where padding is unavoidable—the model may treat spurious regions as meaningful, obscuring true temporal patterns and impairing overall classification performance.

In certain controlled scenarios—such as when zero is a valid and meaningful measurement that naturally appears in the data—padding with zeros may still convey some structural information about sequence length. However, if zero values are frequent or hold no meaningful semantics in a given domain, padding risks merging authentic signals with artificially introduced noise, hindering the network’s ability to differentiate between real observations and placeholders.

These findings highlight the need for careful consideration when applying padding and z-normalization together, especially for sequences of differing lengths. Despite the recognized benefits of z-normalization in accelerating convergence and promoting stable training, practitioners should remain mindful that zero-padding can confound learned feature representations—ultimately hindering the performance of 1D CNN-based TSC models such as InceptionTime [R4].”

"While some modified 1D CNN architectures attempt to handle variable-length inputs through masking layers [R5, R6], these approaches remain limited in scope and are typically evaluated only against older baselines such as DTW. Such masking-based strategies implicitly assume a certain level of temporal correlation between short and long sequences, which can lead to misrepresentations. In contrast, our method avoids these assumptions by mapping each time point into a pixel, allowing each time series to be processed independently without enforced temporal alignment across samples. Additionally, in regions where no actual data points exist, our approach inherently produces an effect equivalent to masking. This ensures that meaningful time steps are emphasized while areas without valid data are effectively ignored—retaining the advantages of masking-based strategies without explicitly relying on them."

·         [R1] Ratanamahatana, Chotirat Ann, and Eamonn Keogh. "Three myths about dynamic time warping data mining." Proceedings of the 2005 SIAM international conference on data mining. Society for Industrial and Applied Mathematics, 2005.

·         [R2] Rakthanmanon, Thanawin, et al. "Addressing big data time series: Mining trillions of time series subsequences under dynamic time warping." ACM Transactions on Knowledge Discovery from Data (TKDD) 7.3 (2013): 1-31.

·         [R3] Bagnall, Anthony, et al. "The great time series classification bake off: a review and experimental evaluation of recent algorithmic advances." Data mining and knowledge discovery 31 (2017): 606-660.

·         [R4] Ismail Fawaz, Hassan, et al. "Inceptiontime: Finding alexnet for time series classification." Data Mining and Knowledge Discovery 34.6 (2020): 1936-1962.

  • [R5]: Sawada, Azusa, et al. "Convolutional neural networks for time-dependent classification of variable-length time series." 2022 International joint conference on neural networks (IJCNN). IEEE, 2022.
  • [R6]: Schneider, Manuel, et al. "An end-to-end machine learning approach with explanation for time series with varying lengths." Neural Computing and Applications 36.13 (2024): 7491-7508.

Comments 2: The author could consider transitions between sections. It could be smoother.

Response 2: We apologize for any lack of coherence in the manuscript’s overall flow, and we sincerely appreciate your guidance on this matter. In response, we have carefully revised our manuscript overall so that the transitions between sections more seamlessly flow throughout the paper.

Comments 3: In Figure 1, it should clearly indicate which graph corresponds to the "left" and "right" positions.

Response 3: We appreciate and apologize for letting them slip through. We have updated the figure and the caption. The modified figure can be found in Section 1. Introduction, Figure 1.

Comments 4: Please add quantitative details about the performance improvements (e.g., accuracy, MAP) achieved with TIP compared to specific baseline methods.

Response 4: Thank you for pointing this out. In response to the reviewer's feedback, we have added quantitative details with three tables regarding the performance improvements achieved with TIP. The tables can be found in Appendix 2.1 (Detailed Results)

Comments 5: The authors mentioned this content: “Through rigorous evaluations against 10 baselines across 11 real-world benchmarks, we show TIP consistently outperforms existing methods, with 2-5% higher accuracy and 10-25% greater macro average precision. Using a straightforward LeNet-like CNN, we confirm that TIP’s gains stem from its data representation, not model complexity.” Is it your objective or your result? It is presented in the introduction section as your contribution.

Response 5: Thank you for pointing out the vagueness in our manuscript. Upon review, we agree that the sentence previously blurred the distinction between the objectives and results of our study. To address this, we have revised the sentence to present it as an objective rather than a result. The updated text is as follows and can be found in Section 1, Introduction:

“Our contributions are twofold:

·         We introduce TIP, a simple yet robust method that converts variable-length time series into a 2D pixel-based format, mapping time steps to the vertical axis and values to the horizontal. This representation enables vision models, including CNNs, to capture key local patterns effectively while preserving data integrity.

·         We validate TIP by employing a straightforward LeNet-like CNN, demonstrating that its performance gains arise from the data representation itself rather than the complexity or power of the vision model.”

Comments 6: Expand on the explanation of the mapping process, particularly how the boundary constants ?−, ?+ are determined. This would make the method more accessible to readers unfamiliar with similar techniques.

Response 6: Thank you for your insightful feedback. In response, we have expanded the explanation of the mapping process and included relevant references to clarify how the boundary constants ?−, ?+ are determined. These boundary constants are essential in mapping time series data into a 2D representation or image format, as they ensure that all time series values are included within the defined range and represented accurately. This ensures a consistent transformation process across various datasets.

  To elaborate further, we have updated the manuscript with the following explanation and have included the references.

“First, we scale the time series values to fit in the image; all time series data for each timestep t ∈ [1, L_n] fit to the range [β−, β+], where β−, β+ are boundary constants. This approach ensures that every time series value is effectively mapped to a corresponding position within the 2D representation, preserving the full range of the original data [R1 ,R2, R3].”

·         [R1] Hatami, Nima, Yann Gavet, and Johan Debayle. "Classification of time-series images using deep convolutional neural networks." Tenth international conference on machine vision (ICMV 2017). Vol. 10696. SPIE, 2018.

·         [R2] Wang, Zhiguang, and Tim Oates. "Encoding time series as images for visual inspection and classification using tiled convolutional neural networks." Workshops at the twenty-ninth AAAI conference on artificial intelligence. 2015.

·         [R3] Xu, Gaowei, et al. "Online fault diagnosis method based on transfer convolutional neural networks." IEEE Transactions on Instrumentation and Measurement 69.2 (2019): 509-520.

Comments 7: Please discuss potential limitations or scenarios where the TIP representation might struggle.

Response 7: Thank you for your valuable feedback. In Section 6 (Conclusion), we have discussed the potential limitations and scenarios where our method might face challenges. For instance, our approach may encounter computational issues when configuring the 2D representation space for extremely long time series, a challenge that remains unresolved across many other TSC methods as well. Additionally, in high-dimensional scenarios, where each variate is treated as a separate image channel, computational efficiency might be affected due to increased sparsity and dimensionality. We have acknowledged these limitations and identified them as areas for future research.

    The corresponding paragraph in the manuscript is as follows:

“This work highlights the challenges inherent in applying current deep learning methods to the variable-length TSC problem. To address these challenges, we propose TIP, a novel approach that directly maps time series data into 2D image representations. We believe our findings contribute significantly to advancing the handling of variable-length time series, which represent the most natural form of time series data encountered in real-world applications.

While our results in variable-length TSC are promising, several issues remain unresolved. One notable concern is the potential for information loss when resizing series, particularly for long sequences—a challenge that is pervasive and yet unresolved in current TSC methods. Furthermore, our method treats each variate as a separate channel, which could pose challenges when dealing with high-dimensional data or sparse datasets. Such scenarios may result in computational inefficiencies or difficulty capturing the complex interdependencies between variates. To address these challenges, future research could explore methods to better preserve interdependencies in multivariate time series. For example, instead of treating each variate as a distinct channel, randomly positioning multivariate images into a shared channel stack could allow the backbone vision model to capture intricate relationships between variables more effectively. Alternatively, incorporating feature extraction or attention mechanisms tailored to multivariate data may enhance both computational efficiency and performance. Resolving these limitations remains an open area for further investigation, which we aim to address in future work.”

We hope this discussion adequately addresses your comment and provides greater transparency about the limitations of our method.

Comments 8: It would be beneficial to compare the 2D-CNN to more advanced architectures to demonstrate the method's robustness.

Response 8: Thank you for your thoughtful suggestion. We also agree that additional analysis is necessary to fully explore TIP’s robustness. While we acknowledge the value of comparing TIP with more advanced architectures, such as state-of-the-art vision models, this falls beyond the scope of the current work. Our primary objective in this study is to validate TIP by employing a straightforward LeNet-like CNN, demonstrating that its performance gains stem from the data representation itself rather than the complexity or power of the vision model. We leave these comparisons and analyses as an avenue for future research, which could further highlight TIP’s adaptability and effectiveness across a wider range of architectures.

Comments 9: Provide more details about the UCR datasets used, especially those with variable-length characteristics.

Response 9: We appreciate your valuable feedback. In response, we have provided additional details about the UCR datasets used in our study, with a particular focus on those with variable-length characteristics. These details have been included in the Appendix A.1 Datasets Description:

“The datasets used in this study are from the UCR Time Series Archive[R1], a widely utilized resource in time series research. Originally introduced in 2002 with 16 datasets, the archive has grown to encompass 128 datasets, making it a critical benchmark in the field. Among these, 11 datasets are specifically designed for variable-length time series, presenting unique challenges for classification methods. A detailed description of these datasets can be found in Table A1.

One example of such a dataset is GesturePebble, which showcases the variability in time series lengths. The data is derived from the z-axis readings of a 3-axis accelerometer in Pebble smartwatches, worn on participants’ wrists as they perform six distinct gestures. The dataset consists of 304 gestures, with lengths varying naturally due to differences in individual movements. Two versions of the dataset were created:

·         GesturePebbleZ1: The training set contains data from all participants in the first session, while the test set contains data from the second session.

·         GesturePebbleZ2: Participants are split between the training and testing sets, making it more challenging due to the distinctiveness of individual gaits and movements.”

[R1] Dau, Hoang Anh, et al. "The UCR time series archive." IEEE/CAA Journal of Automatica Sinica 6.6 (2019): 1293-1305.

Comments 10: Please expand on why TIP performs comparably or better than InceptionTime in some cases, particularly for MAP scores. Are there specific characteristics of the datasets that favor TIP?

Response 10: Thank you for your insightful comment. We acknowledge that understanding the precise reasons why TIP outperforms models like InceptionTime in some cases and in terms of MAP scores in certain cases is a complex matter, and it may stem from several interacting factors.

   One possible explanation is that TIP's data representation method captures key local patterns in the original time series more effectively without introducing distortions from truncation or interpolation. This approach likely preserves the integrity of temporal dynamics, which could be particularly beneficial for datasets with high variability or sparse but informative patterns.

   Another factor could be the reliance of InceptionTime and similar models on complex preprocessing steps and model architectures, such as ensemble techniques and inception blocks. These elements, while powerful, may inadvertently amplify noise or irrelevant features in datasets where critical information lies in subtle, localized patterns—an area where TIP’s 2D-CNN backbone may excel due to its focus on spatially coherent representations.

   We included this analysis into our manuscript in Section 4.4 (Results). However, we acknowledge that this is a hypothesis and requires further analysis to validate. Future work could involve a deeper investigation into the dataset characteristics, such as class imbalance, feature sparsity, or variability in sequence length, to identify patterns that favor TIP over other methods. We appreciate the opportunity to explore this in more depth as part of ongoing research.

“Lastly, 1D-CNN-based models generally performed well; however, our method, TIP, outperformed them across most datasets. Notably, InceptionTime, which employs truncation as a preprocessing step, achieved the lowest performance despite being the latest and ostensibly the best-performing algorithm in its category. This result indicates that truncation as a preprocessing method significantly distorts the original data information. When padding, instead of truncation, was applied, InceptionTime, ResNet, and FCN showed progressively higher performance. This improvement can be attributed to their complex structures: InceptionTime utilizes ensemble techniques and inception blocks, which are modern feature extraction methods; ResNet incorporates residual connections; and FCN consists solely of convolutional layers. Additionally, padding generally outperformed interpolation, likely because interpolation excessively alters the temporal dynamics of the original data. However, overall, these imputation methods using artificial values resulted in lower performance compared to our method, which preserves data integrity by employing a straightforward vision model with a 2D-CNN backbone.”

Comments 11: Please elaborate on how TIP could be adapted or extended for multivariate time series with complex interdependencies.

Response 11: We appreciate the valuable feedback and have updated our conclusion to address the adaptation and extension of TIP for multivariate time series with complex interdependencies. Below is the revised text in Section 6. Conclusion:

“This work highlights the challenges inherent in applying current deep learning methods to the variable-length TSC problem. To address these challenges, we propose TIP, a novel approach that directly maps time series data into 2D image representations. We believe our findings contribute significantly to advancing the handling of variable-length time series, which represent the most natural form of time series data encountered in real-world applications.

While our results in variable-length TSC are promising, several issues remain unresolved. One notable concern is the potential for information loss when resizing series, particularly for long sequences—a challenge that is pervasive and yet unresolved in current TSC methods. Furthermore, our method treats each variate as a separate channel, which could pose challenges when dealing with high-dimensional data or sparse datasets. Such scenarios may result in computational inefficiencies or difficulty capturing the complex interdependencies between variates. To address these challenges, future research could explore methods to better preserve interdependencies in multivariate time series. For example, instead of treating each variate as a distinct channel, randomly positioning multivariate images into a shared channel stack could allow the backbone vision model to capture intricate relationships between variables more effectively. Alternatively, incorporating feature extraction or attention mechanisms tailored to multivariate data may enhance both computational efficiency and performance. Resolving these limitations remains an open area for further investigation, which we aim to address in future work.”

Round 2

Reviewer 2 Report

Comments and Suggestions for Authors

The authors have solved my concerns and the paper is ready for publication.

Author Response

Dear Reviewer 2,

Thank you for your insightful reviews through the process. We greatly appreciate your time and effort in evaluating our manuscript.

Best regards,

Reviewer 3 Report

Comments and Suggestions for Authors

In the Experiment section, the experiment should include more qualitative results, such as visualizations of the TIP-transformed data and its effect on classification performance. Please also discuss the statistical significance of the results to reinforce the reliability of the findings.

In the Discussion section, please elaborate on the potential limitations of TIP, such as its scalability to extremely large datasets or multivariate settings.

For Figure 4, the characters are too small to read, and the image itself is too small. Please enlarge and clarify the image for better readability.

Author Response

Thank you very much for taking the time to review this manuscript. Below, we provide detailed responses to your comments and the corresponding revisions/corrections highlighted/in track changes in the re-submitted files. We hope our responses clarify and elaborate on the points raised.

Comments 1: In the Experiment section, the experiment should include more qualitative results, such as visualizations of the TIP-transformed data and its effect on classification performance. Please also discuss the statistical significance of the results to reinforce the reliability of the findings.

Response 1: The authors acknowledge the reviewer's comment on the necessity of qualitative visualization of the experimental results. To address this, the explanation has been provided in Figure 4. This study primarily explores the potential application of 2D-CNN techniques to TIP-transformed data, as demonstrated in Tables 2 and 3. While statistically significant results were identified during the research process, the limited two-day revision period constrained the authors from re-executing and reorganizing the algorithm comprehensively. Instead, the focus of this paper is on comparing the application results of the proposed algorithm with those of existing approaches, highlighting the significance of this study.

Comments 2: In the Discussion section, please elaborate on the potential limitations of TIP, such as its scalability to extremely large datasets or multivariate settings.

Response 2: We appreciate the reviewer’s valuable feedback on this point. In response, we have elaborated the potential limitations of TIP, such as scalability or computational challenges, for extremely high multivariate settings. These revisions can be found in Section 5.1 (What about in multivariate setting?).

“As shown in Table 1, our method achieves performance comparable to other methods, despite lacking features specifically designed to capture spatial patterns between variables. While InceptionTime, the current state-of-the-art TSC integrated with length-uniformization techniques, achieved the highest accuracy, our method delivered a comparable accuracy score and outperformed it in MAP. Nevertheless, our approach may face computational challenges when scaling to extremely large datasets or handling high-dimensional multivariate settings. This is primarily because each variate is treated as a separate channel and stacked, which could increase memory and processing demands. However, our results suggest that even under these constraints, the method remains effective in maintaining high performance and robustness for complex multivariate TSC problem.”

Comments 3: For Figure 4, the characters are too small to read, and the image itself is too small. Please enlarge and clarify the image for better readability.

Response 3: We appreciate and apologize for letting them slip through. We have updated the figure; We enlarged the overall fonts and plots. The modified figure can be found in Section 4. Experiment, Figure 4, page. 9 of 17.
